# China's Wealth Capital Stock Mapping via Machine Learning Methods

**Lulu Ren** , **Feixiang Li, Bairu Chen, Qian Chen** , **Guanqiong Ye and Xuchao Yang** *

Ocean College, Zhejiang University, Zhoushan 316021, China
* Correspondence: yangxuchao@zju.edu.cn

**Abstract:** The frequent occurrence of extreme weather and the development of urbanization have led to the continuously worsening climate-related disaster losses. Socioeconomic exposure is crucial in disaster risk assessment. Social assets at risk mainly include the buildings, the machinery and the equipment, and the infrastructure. In this study, the wealth capital stock (WKS) was selected as an indicator for measuring social wealth. However, the existing WKS estimates have not been gridded accurately, thereby limiting further disaster assessment. Hence, the multisource remote sensing and the POI data were used to disaggregate the 2012 prefecture-level WKS data into 1000 m × 1000 m grids. Subsequently, ensemble models were built via the stacking method. The performance of the ensemble models was verified by evaluating and comparing the three base models with the stacking model. The stacking model attained more robust prediction results (RMSE = 0.34, $R^2$ = 0.9025), and its prediction spatially presented a realistic asset distribution. The 1000 m × 1000 m WKS gridded data produced by this research offer a more reasonable and accurate socioeconomic exposure map compared with existing ones, thereby providing an important bibliography for disaster assessment. This study may also be adopted by the ensemble learning models in refining the spatialization of the socioeconomic data.

**Keywords:** WKS; economic exposure; ensemble model; random forest; point-of-interest





## 1. Introduction

As an important indicator of the economic status of a country (or region), the gross domestic product (GDP) is widely used as a landmark indicator of the economic exposure in disaster assessment. However, from the perspective of statistical processes, the GDP only contains the final outcome of a country's (or region's) productive activities over a period of time at the current market prices. In other words, the GDP can only measure the market value of the final products over a period (a "flow" indicator). By contrast, the wealth capital stock (WKS), also called "net" capital stock, can measure the value of the capital stock, that is, the (hypothetical) value that could be achieved if the complete fixed stock were to be sold at today's market prices [1]. Such fixed assets generally include the buildings' assets, the tools, the machinery, and the equipment and similar infrastructure. In addition, the evolution of the capital over time is controlled by the flow of investment and depreciation (i.e., the consumption of the fixed capital). Thus, we believe that the asset stocks as a "stock" indicator is more consistent with real socioeconomic exposure, a theory that has been increasingly supported by the recent research. Previously, Seifert et al. [2] mapped the distribution of the industrial and the commercial values in Germany to estimate the risk of future flooding along the Rhine River. The 2013 Global Disaster Risk Assessment report also stated that capital stocks had been used to replace the GDP as a proxy for the economic risk to natural disasters [3]. Holz et al. [1] followed the latest OECD methodology to calculate the long-term series of the provincial and the national capital estimates in China by distinguishing WKS from the capital services.

Although China does not have any official data on its capital stock, estimations at different administrative scales have been produced by many researchers, some of whom have presented accurate results at the prefecture level. This development allows for the WKS dataset to be used for more accurate disaster risk management. In the past, Wu et al. estimated the WKS of 344 prefectural cities in China and used it to assess the direct economic losses from earthquake disasters.

Similar to the GDP, the WKS data based on the administrative divisions contain limitations, such as the failure to consider the uneven distribution of residents within the administrative divisions, the aggregation of socioeconomic activities, and the difference in the social wealth distribution, which leads to rough and impractical disaster risk assessment results. The grid data can effectively address these shortcomings by depicting the WKS density distribution within the administrative districts. In addition, the grid data provide the basis for flexible data fusion to facilitate further spatial analysis and calculation. Many studies about the downscaling of the socioeconomic and the demographic data have been conducted in recent years. Remote sensing data, such as the nighttime light (NTL) images and the vegetation indices, are widely used in the spatialization of the social and the economic data, including the GDP and the electricity consumption [4]. This scenario shows the strong correlation between the multisource remote sensing data and the socioeconomic data. In regard to the model construction, Wu et al. [5] attempted to grid the WKS data via the auxiliary data decomposition method, but the gridding process requires linear decomposition, and the results cannot be accurately estimated because of the complexity of human economic activities. By considering the complexity of the social wealth distribution, the machine learning algorithms represented by the random forest (RF) have been proven to perform well in the spatial model construction [4,6,7]. Points of interest (POIs), a type of social-sensing big data that are closely related to human activities, are increasingly used to assist in the identification of the urban functional areas because they can clearly describe the type and the location of human activities [8–13]. POI data have a great potential in the study of the social and the economic characteristics, especially in complex urban areas [14–16]. However, the training methods, the learning rules, and the objective functions tend to vary greatly because each model has its own characteristics for certain datasets. The multimodel ensemble refers to the method of combining different single models, thereby providing more reliable and robust simulation results than would be achieved by simply using a single model. Therefore, we introduced three different types of models, namely, the RF, the cubist, and the extreme gradient boosting (XGBoost) models, and integrated them via stacking.

By considering the diversity of the various regions in China, the POIs with multisource remote sensing data were combined in an ensemble learning model prior to disaggregating the 2012 prefecture-level WKS data into 1000 m × 1000 m grids. Previous data experiments have confirmed the effectiveness of the stacking model.

## 2. Data and Methodology

### 2.1. Data Collection and Preprocessing

#### 2.1.1. WKS Data

The WKS dataset of China's 344 prefectures was obtained from the study of Wu et al. (2012), who used the perpetual inventory method to estimate the WKS of the 2012 prefecture-level cities in mainland China [17].

#### 2.1.2. Remote Sensing Datasets

a.    NTL data

The global satellite-derived radiance-calibrated NTL product for 2012 at a 1 km resolution from the NPP-VIIRS was downloaded from the Earth Observation Group of Colorado School of Mines (https://eogdata.mines.edu/nighttime_light/annual/v20/, accessed on 1 March 2022). The VNL_v20 product provides the global VIIRS night light annual time series with the annual continuous processing from the monthly cloudless average radiation grid in 2012, which can be used to solve most of the light saturation problems. Previous

studies have confirmed its higher simulation accuracy compared with DMSP/OLS; hence, NPP/VIIRS is considered highly suitable for simulating the regional economies [18–21].

b.    Vegetation index

The normalized difference vegetation index (NDVI) dataset from the Satellite Pour I 'Observation de la Terre (SPOT) was acquired from the Vlaamse Instelling Voor Technologisch Onderzoek. The SPOT S10 product provides land surface reflectance at 1000 m resolution from seven spectral bands every 10 days. The annual maximum NDVI (NDVI$_{MAX}$) images are calculated using the maximum value composite method (to eliminate the effect of cloud contamination) and the bidirectional reflectance distribution function [22].

$$NDVI_{MAX} = MAX(NDVI_1, NDVI_2, \ldots, NDVI_{36}) \tag{1}$$

where the subscripts 1 to 36 represent the 36 10-day SPOT NDVI images in 2012.

c.    Land surface temperature (LST) data

The LST dataset was obtained from the MODIS MOD11A2 product (https://ladsweb.modaps.eosdis.nasa.gov/search/, accessed on 1 March 2022), which is the eight-day average of the daily MOD11A1 LST product. The average measurements at daytime and nighttime are stored separately. Here, the mean daytime and nighttime LSTs (i.e., LST-day and LST-night, respectively) for 2012 were calculated.

d.    Digital elevation model (DEM) data

The original DEM data were obtained from the ASTER GDEM Version 2 dataset of the Earth Remote Sensing Data Analysis Center of Japan (http://www.gdem.aster.ersdac.or.jp/search.jsp, accessed on 1 March 2022), with a spatial resolution of 30 m. The DEM data were projected into the Albers Conical Equal Area projection and then resampled to a new dataset with a pixel size of 1000 m via bilinear interpolation. The elevation and slope raster layers were generated from the new image in ArcGIS 10.2.

### 2.1.3. Road Network Data

The road network vector dataset was obtained from the Data Center for Resources and Environmental Sciences, Chinese Academy of Sciences (http://www.resdc.cn/). This dataset includes the national highways, the railways, the provincial highways, the county highways, the township-level roads, the city roads, and others. The vector data were used to calculate the corresponding raster data layer of the nearest road density (e.g., DtC-road, road length per unit of area) at 1 km resolution in ArcGIS 10.2.

### 2.1.4. POI Data

The POIs of mainland China in 2010 were derived from a mobile map service platform, namely, Baidu Map (http://map.baidu.com). A total of 5,152,950 POI records [14] were obtained, with each POI record typically containing the point's location, the name, the category, and the attributes. The Baidu POIs dataset also records the above information in Chinese phrases and classifies the points [23]. According to the application purpose and actual requirements of this study, we selected a top-level category of 20 labels that contained educational facilities, entertainment services, and residential communities because they could provide a good overview of the functions of the geographical entities that are closely related to human life.

Kernel density estimation (KDE) was used to represent the spatial influence region of the POIs because it could determine the location influence of the first law of geography and it is superior to other density expression methods. Bandwidth is the most important control parameter of the KDE method, but many studies have established the absence of an accurate algorithm to determine its optimal value [24]. In this study, we calculated the density grid from 500 to 8000 m at intervals of 500 m and selected 2000 m as the bandwidth in accordance with the out-of-bag (OOB) error minimization principle of RF [14]. Subsequently, we

generated continuous and smooth kernel density surfaces for each category by using the kernel density tool in ArcGIS10.2.

Given that the selected POI data consisted of 20 layers, we introduced the principal component analysis (PCA) to reduce the dimension of the input data during modeling while avoiding the massive calculation of unusable data. The PCA is a data analytic method that is often used for the dimensionality reduction of high-dimensional data and that can be used to extract the main feature components of the data. According to the maximum separability for the PCA, the variance of the first principal component would exceed 90%. Therefore, only the first principal component was selected to output the composite density surface (POIs-den) raster with a resolution of 1000 m.

*2.2. Methodology*

The spatialization approach involves four main steps (Figure 1): (1) model learning and stacking; (2) dasymetric mapping; and (3) validating the predictive ability of the models.

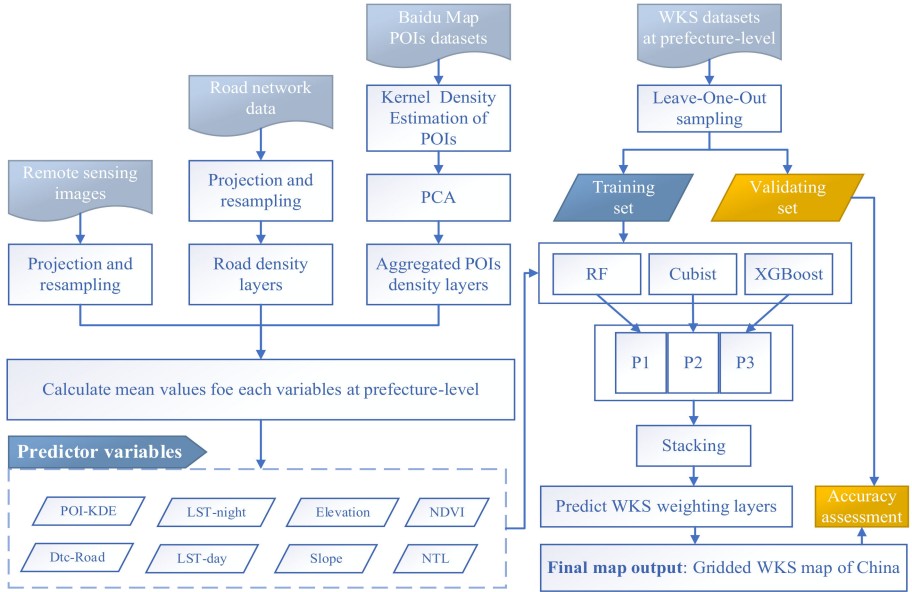

**Figure 1.** Flowchart of the proposed methodology for WKS estimation.

2.2.1. Building the Base Models

In circumstances where the size of the dataset is small and the final results are subject to a large number of factors (i.e., the circumstances of this study), the use of different base models and the stacking method is considered the appropriate choice. Here, the RF, the cubist, and the XGBoost models were used as the base models.

The RF algorithms are classic machine learning methods for nonlinear and nonparametric work. This machine learning method has strong adaptability to problems that involve tight coupling data, which means that it can reflect the non-distinct interaction among data features [25]. The RF method can obtain appropriate results by using only a few parameters, and as such is different from other congeneric algorithms that require many parameters to avoid the outliers, the noise, and the overfitting problems [25,26]. The RF algorithms also provide a special OOB error for the unbiased estimation of the model error. The OOB error can be generated from the error of approximately one-third of the data, which otherwise could not be extracted randomly when training a tree. By calculating the average OOB error of each tree, the prediction error of the whole model can be evaluated [27]. The RF models generally reduce the voting weight of trees with larger OOB errors to improve the prediction accuracy of the model.

The cubist models are a type of prediction-oriented regression model, which is a rule-based extension of Quinlan's M5 model tree. In cubist modeling, the linear regression model is used recursively at the terminal node, and the predicted value of the previous node

is considered for smoothing. The tree is reduced to a set of rules, which are initially taken as paths from the top of the tree to the bottom. The rules are eliminated by pruning and/or combined for simplification. The final prediction is a simple average of the predictions from each model tree [26]. Such predictions have been shown to outperform the outputs of the linear regression models. The running time of cubist models is also much shorter than those of the classification and the regression tree methods [28].

The XGBoost models are distributed machine learning algorithms based on the boosting learning framework [29]. The core objective of the XGBoost models is to minimize the model error by using the gradient descent algorithm. In the training process, all samples are learned in each round of the training, and every sample has the same weight in the initial state. Subsequent learners learn from the residual of the previous results, and the new learner is established in the gradient descent direction of the loss function in previous learners. The XGBoost uses the weighted fusion method to consider the weighted average of the results of each tree as the final output [30], which improves the accuracy of the model effectively.

In training the base models in this study, eight raster layers (e.g., NTL brightness, NDVI, LST-day, LST-night, slope, elevation, Dtc-road, and POI density) were aggregated at the ground level as the independent variables. The natural logarithm of the WKS density was used as the dependent variable. Then, the three models (e.g., the RF, the cubist, and the XGBoost models) were used to construct the relationship between the geographic indicators and the WKS density. The raster layers were inputted into the constructed model, and then the prediction layers were calculated. By using the prediction layer as pixel-level distribution weights, the prefecture-level WKS density data could be decomposed into 1000 m resolution grids as follows:

$$\mathrm{WKS_{grid}} = \frac{\mathrm{WKS_{prefecture}} \times \mathrm{W_{grid}}}{\mathrm{W_{prefecture}}} \tag{2}$$

where $\mathrm{W_{grid}}$ is the WKS distribution weight of a 1000 m $\times$ 1000 m grid area, $\mathrm{W_{prefecture}}$ is the sum of the total WKS distribution weights for a prefecture, $\mathrm{WKS_{prefecture}}$ represents the total WKS of a county, and $\mathrm{WKS_{grid}}$ is the gridded WKS distribution.

### 2.2.2. Ensemble Learning and Model Fitting

The commonly used ensemble learning methods are stacking, bagging, and boosting. Bagging and boosting methods are often used to combine homogeneous models, whereas stacking is often used to combine different trained models.

After the base model training, multiple linear regression was chosen for model stacking, with the output of the base model selected as the independent variable and the actual result as the dependent variable. In addition, the stacking effect was verified using the reserved validation set data. The modeling was implemented as follows:

$$\mathrm{P_s} = \mathrm{a_0} + \mathrm{a_1} \times \mathrm{P_{rf}} + \mathrm{a_2} \times \mathrm{P_{cb}} + \mathrm{a_3} \times \mathrm{P_{xg}} \tag{3}$$

where $\mathrm{P_s}$ is the predicted value of the integrated model; $\mathrm{P_{rf}}$, $\mathrm{P_{cb}}$, and $\mathrm{P_{xg}}$ are the predicted values of the RF, the cubist, and the XGBoost models, respectively; and $\mathrm{a_i}$ is the linear regression coefficient, which can be determined via the least-square method.

### 2.2.3. Dasymetric WKS Mapping

Dasymetric mapping methods can ensure constant totals within a given administrative division [31] and reveal highly reliable patterns of spatial distribution in the population and the GDP spatialization [32,33]. In this study, we initially used the water body mask to obtain the inhabited area, and on this basis, the WKS density distribution was further

divided. Then, we took the $W_{prefecture}$ (Section 2.2.2) as the weight layer and redistributed the prefecture level as follows:

$$WKS^{ij} = WKS^j \times WKS^i_{grid} \qquad (4)$$

where j represents the index of the prefecture from 1 to 344, i represents the index of a grid in the j-th prefecture, $WKS^{ij}$ represents the allocated WKS of the i-th grid in j-th prefecture, $WKS^j$ represents the total WKS in the j-th prefecture, and $WKS_{grid}$ represents the weight cost of the i-th grid in Equation (2).

### 2.2.4. Accuracy Validation

We selected the leave-one-out (LOO) method for cross validation to verify the accuracy of the WKS maps from the individual basic learners and the multiple models. The LOO method is widely used in cross validation because it can evaluate the performance of a model with a limited dataset quickly and accurately. Here, we took 90% of the samples as the training set and the remaining 10% as the validation set. Consistency in data features in the training set and validation set were ensured by using the Fisher-Yates shuffle algorithm to arrange the data randomly. Subsequently, the top 10% of the data was extracted as the validation set, whereas the remainder was used as the training set.

$$e_{RE} = \frac{|y - \bar{y}|}{\bar{y}} \qquad (5)$$

$$e_{MRE} = \frac{\sum(e_{RE})}{n - 1}, \qquad (6)$$

$$e_{RMSE} = \sqrt{\frac{\sum(y - \bar{y})^2}{n - 1}}, \qquad (7)$$

Equations (5)–(7) were used to evaluate the accuracies of the prediction datasets. The MRE was used to reflect the degree to which the model output was close to the real result. The MRE is a dimensionless evaluation indicator used for depicting the accuracy of a model. In contrast to the MRE, the RMSE, as a convincing indicator for evaluating models, is often used to compare the accuracies of different models based on the dimension. However, the accuracy verification can only determine the error of the average density of the prefecture-level administrative units, but it cannot directly obtain the simulation accuracy of the actual WKS distribution characteristics. Therefore, two additional steps were performed: (1) generate gridded WKS maps and (2) compare the performance of the different models in the gridded WKS maps. We selected four typical regions of China for the detailed analysis and comparison of the gridded WKS maps.

## 3. Results

### 3.1. Accuracy Assessment

Table 1 shows the accuracy evaluation results of the four models. The XGBoost outperformed the two other base models, and the performances of the RF and the cubist were analogous. This outcome can be explained by the fact that the RF and the cubist models require linear classifiers, which often achieve good classification results when using large sample sizes. By contrast, the flexible classifier selection of the XGBoost is more apparent when the dataset size is small. The stacking model also showed good performance, especially because the WKS data varied greatly in the different numerical ranges. Furthermore, the stacking model utilizes a more flexible and extensive selection and combination of base classifiers [15,34,35]. In conclusion, the ensemble model for the WKS prediction (i.e., the stacking model) supersedes the three other base models despite the limitations of the dataset size and the considerable differences in the spatial distribution of each sample because of geographic, economic, and political factors.

**Table 1.** Accuracy evaluation of estimated WKS maps.

|  | RF | Cubist | XGBoost | Stacking |
|---|---|---|---|---|
| MRE | 4.183% | 4.248% | 4.044% | 3.798% |
| RMSE | 0.371293 | 0.374573 | 0.361901 | 0.337656 |

Figure 2 shows a scatter plot of the predicted and the true values of the models. A perfect model is represented by a straight line (y = x). The degree of deviation of the scattered points from the straight line can reflect the effect of the model. Firstly, in the low-value region (0–7), the results of all the models were the least desirable (i.e., widely scattered), which may be explained by the scarcity of data. The performances of all the models in this range were consistent with one another, which also indicated that the data were significantly different from the training data. Secondly, in the middle range (7–9), most of the data were concentrated, and the performance of each model varied. The values of the RF model were more dispersed and located farthest from the y = x line, those of the cubist model were more dispersed along the y = x line, and those of the XGBoost model were below the y = x line. Meanwhile, the values of the stacking model were highly concentrated near the y = x line. Thirdly, in the high-value region (9–12), the predicted values of the RF and the XGBoost models were relatively small, as shown in the figure where the scatter points are located below the line y = x, while the points of the cubist model were mostly located on the line. The prediction results of the stacking model are shown as adopting the prediction results of the cubist model, that is, most of the scattered points are located on the line. Therefore, the stacking model can better balance the results of the based models. The low performance of the stacking model in the low-value region can be explained by the poor data regularity, but this issue is likely to affect all models.

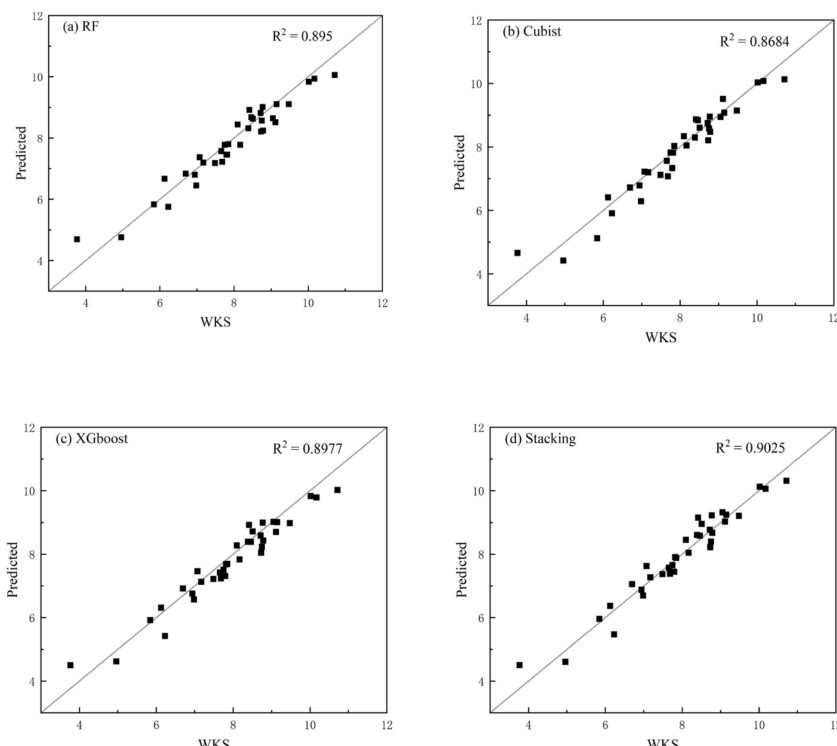

**Figure 2.** Scatter plot of the predicted and true values of (**a**) RF, (**b**) cubist, (**c**) XGBoost, and (**d**) stacking models. (note: predicted and true values of the models are both logarithmic).

### 3.2. Gridded WKS Maps and Model Comparison

By implementing the methods discussed in Section 2.2.3, we generated a WKS gridded density map at a resolution of 1000 m for the RF, the cubist, the XGBoost, and the stacking models. The final gridded density and the final fusion map results are shown in Figure 3d. The spatial distribution trends of the four models were similar for mainland China. In particular, high-value regions were mainly distributed in the metropolitan areas of the Yangtze River Delta, Pearl River Delta, Chengdu–Chongqing, and Beijing-Tianjin. Unsurprisingly, China's four municipalities (e.g., Chongqing, Shanghai, Beijing, and Tianjin) showed the highest stock of fixed assets. The second-highest value region was distributed in other coastal and developed provincial capitals. As depicted by the WKS density map of mainland China in 2012, most of the WKS was mainly distributed in areas with high urbanization levels and relatively complete infrastructure construction.

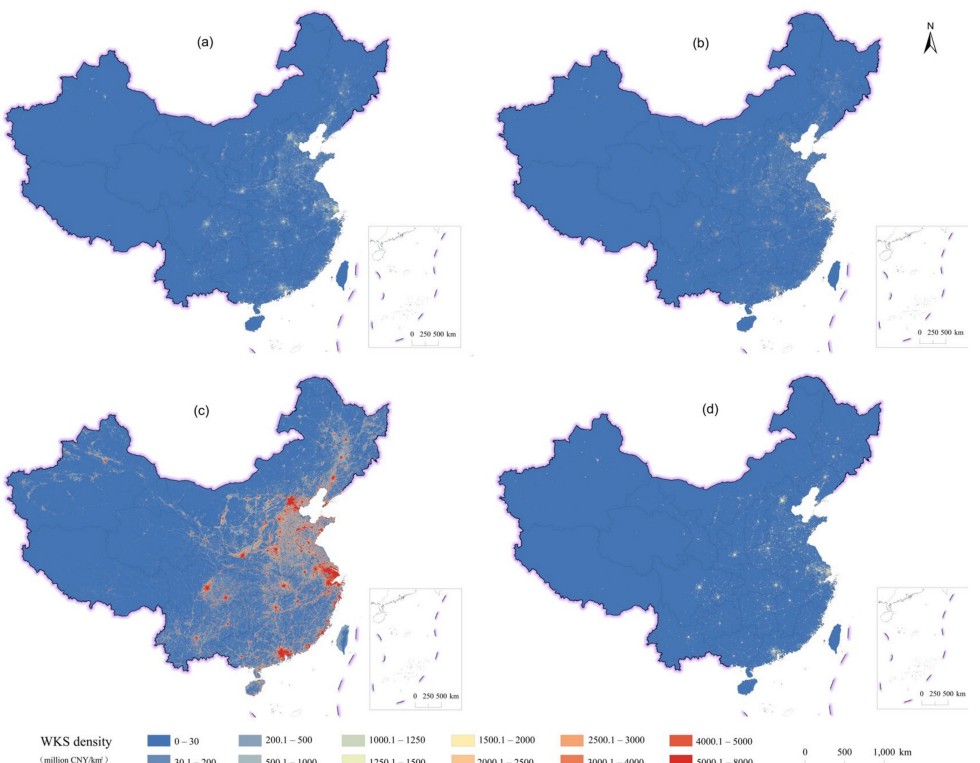

**Figure 3.** Gridded WKS maps in 2012 by using (**a**) RF, (**b**) cubist, (**c**) XGBoost, and (**d**) stacking models.

Next, we used the natural logarithm of the WKS density at the prefecture level as the dependent variable to obtain the numerical range of the four models. Table 2 shows the numerical ranges of the four models on the training data. The natural logarithm values range from 1.495 to 11.647. Correspondingly, the pixel value range of the weight layer, as predicted by the RF model, was the smallest, ranging from 2.210 to 7.944. This finding could be attributed to the predicted value of the RF model always being within the training data range, and the numerical range after voting would be more concentrated. Meanwhile, as the cubist model entails excellent extrapolation, the numerical range was the widest, ranging from 0.181 to 8.961, even outstripping the range of the training set. It also showed an extremely obvious data overflow, especially at the lower boundary. Figure 2 shows that the cubist model had the worst prediction effect in the low value area. The reason for this may be because the cubist model is sensitive to small training data samples, so the prediction effect is poor. Similar to the RF model, the XGBoost model is also a conservative model, and its predicted value will not exceed the training value range (prediction range: 1.777 to 8.820). However, in contrast to the RF model, which pays more attention to the variance, the XGBoost model focuses more on the deviation, hence its larger range. The stacking model, whose data range is from 1.213 to 8.550, integrates the different characteristics of the

base models and can overcome their inherent shortcomings. This feature of the stacking model implies its ability to maximize the model accuracy for achieving a higher quality population fitting.

**Table 2.** Range comparison between the training set and the four models.

|  | Train | RF | Cubist | XGBoost | Stacking |
|---|---|---|---|---|---|
| Lower | 1.495 | 2.210 | 0.181 | 1.777 | 1.213 |
| Upper | 11.647 | 7.944 | 8.961 | 8.820 | 8.550 |

Finally, we compared the differences of the four models by using the four selected key areas of Beijing–Tianjin, Yangtze River Delta, Pearl River Delta, and Chengdu. Figure 4 shows the apparent variations in the modeling results of the selected regions. The maximum of the RF model was less than 1100 million CNY/km$^2$, which was lower than those of the other models, and with a concentrated value range. The large number of warm-colored areas of the RF model surpassed those of the other models, as depicted by the disorderly sprawl. Meanwhile, the cubist model was characterized by fewer warm-colored areas, but the center of each area had a highlighted area, which was especially prominent for the Yangtze River Delta and Pearl River Delta. The results of the XGBoost model were closer to those of the stacking model, which was also consistent with the conclusion presented in Section 3.1. The overall performance of the XGBoost model was depicted by its relatively concentrated data, which was apparent in areas with multiple economic centers, such as the Yangtze River Delta and Pearl River Delta. The performance of the stacking model neutralized the characteristics of the three abovementioned models. The high-value areas were more concentrated, and the megacities (Beijing, Shanghai, Guangzhou, and Shenzhen) were more distinguishable from the surrounding urban agglomerations, which is consistent with the actual distribution. In addition, the predicted value of the ensemble learning model essentially obeys the law in which the WKS is 2.5 times the GDP, as depicted by the characteristic distribution along the roads in the outer edge of the economic zone, which is also in accordance with the main road having a high capital stock value. The aforementioned findings jointly demonstrate the credible predictions of this study.

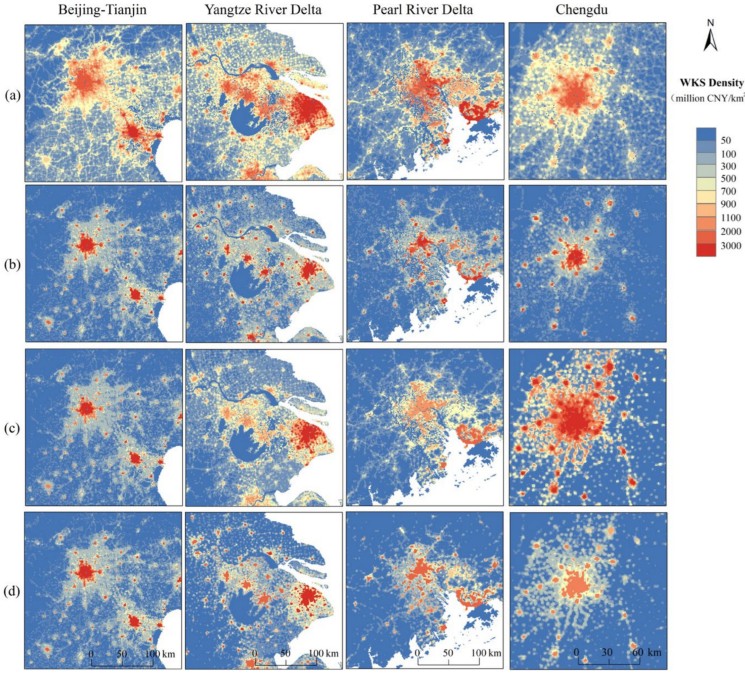

**Figure 4.** Comparison of the WKS maps by using the (**a**) RF, (**b**) cubist, (**c**) XGBoost, and (**d**) stacking models for four metropolitan agglomerations in China.

## 4. Discussion

The WKS data offer a unique perspective for estimating and mapping economic exposures across a range of spatial scales. The main limitation of the previous studies is their overdependence on the GDP data and the spatial modeling method selection. As a widely used economic indicator, the GDP can indeed provide an effective reference for economic activity. However, with the deepening study of disaster assessment, some serious disasters, such as the earthquake in Haiti [36], were found to cause economic losses at levels higher than the GDP of the affected area, thereby rendering the GDP an ineffective indicator. Consequently, searching for more precise indicators of the total physical fixed assets at risk is crucial in disaster risk management. Fixed assets generally include the buildings' own assets, the tools, the machinery and the equipment, and similar infrastructure; thus, it is a more precise indicator of social wealth at risk. In addition, the depreciation and loss of the stock assets can also be measured by the WKS. Recent research on the WKS has demonstrated the great potential of this index in studying various disasters, such as earthquakes [17] and floods [37]. In the spatialization results of this study, we found the evident distribution characteristics of fixed assets, such as high-value areas concentrated in parts with high urbanization levels and perfect infrastructure. The distribution along road shapes outside the economic zone was highly consistent with the high capital stock of the trunk roads. Moreover, the WKS value in the non-high value area was 2.5 times the value of the GDP, which is consistent with the findings in economic research.

Instead of traditional linear modeling methods, an ensemble machine learning algorithm was used in this study to build a complex nonlinear model between the WKS destiny and each indicator factor, thereby achieving improved performance. Previously, Wu et al. [5] used linear decomposition to spatialize the WKS, a widely used spatialization method, by assuming that a linear relationship exists between the fixed assets and certain indicators. The fixed assets and other economic indicators at the prefecture level can also be used to solve this linear relationship, after which the weight is calculated according to the grid-level economic indicators.

The findings of this study offer numerous insights. Firstly, researchers need to assume the existence of a linear relationship between the fixed assets and other economic indicators empirically, which may require several attempts to verify. Secondly, the assumption that this relationship is consistent across the country should be avoided because it is clearly inappropriate. Finally, equal weight must be given to each prefecture-level sample, and the features of special numerical samples may be eliminated. Machine learning methods can overcome existing shortcomings by representing the features within the model without making empirical assumptions. Moreover, special samples can be processed to obtain a model with stronger generalization ability. Existing single-model machine learning models are affected by differences in their internal structures, the training methods used, and their objective functions [4,7], and vary in handling the characteristics of samples with unique features. As discussed in Chapter 3, the stacking model can provide suitable weights to each model at different intervals and obtain better overall results.

Despite the significant improvements in methodology, certain limitations were still apparent in the current study. Given the lack of the grid-scale reference data, we did not evaluate the model results at the grid scale level accurately. Instead, we only compared the model effects by analyzing the data distribution of the validation set. Conversely, the POI data generally lacked information about the range and the value of the POI, such as the highways and the rural roads. Equal weighting was used in the model while ignoring their vastly different WKS values. As for the social-sensing big data, the number of POIs could represent the value of the whole system but only to a certain extent. The existing POI data for rural areas are incomplete, which may underestimate asset stocks in the rural areas of the WKS grid map. However, the asset stocks in rural areas are much lower than those in urban areas. Therefore, the spatial heterogeneity of the POI data in rural areas would not cause significant errors in the spatial results.

## 5. Conclusions

Multisource remote sensing images and POI data were used to develop four machine learning models, namely, the RF, the cubist, the XGBoost, and the stacking models. Subsequently, the built models were used to disaggregate the 2012 prefecture-level WKS dataset in mainland China. A set of spatial distribution data of the WKS in the Chinese mainland in 2012 with a resolution of 1000 m was established. The dataset that represented the social wealth stock revealed the distribution of social wealth in mainland China, which could provide a grid-scale reference for social economic exposure and vulnerability in disaster assessment. Considerable differences were apparent in the economic development and infrastructure levels in China. The map generated in this research also reflected the obvious regional differences in the WKS distribution. The multisource remote sensing data and the social-sensing big data captured economic and social characteristics accurately. Compared with single-machine learning methods, the stacking ensemble model addressed the shortcomings of each of the three primary models more effectively and even showed a better prediction effect. This finding indicates the great potential for the use of more advanced machine learning methods in spatial applications, thereby allowing for more refined socioeconomic factor estimation research in the future.

**Author Contributions:** Conceptualization, X.Y. and L.R.; methodology, X.Y., Q.C. and L.R.; software, L.R. and F.L.; validation, L.R.; formal analysis, L.R.; investigation, L.R.; resources, L.R.; data curation, L.R.; writing—original draft preparation, L.R.; writing—review and editing X.Y., G.Y. and B.C.; visualization, L.R.; supervision, X.Y. and G.Y.; project administration, X.Y.; funding acquisition, X.Y. All authors have read and agreed to the published version of the manuscript.

**Funding:** This work was supported by the Second Tibetan Plateau Scientific Expedition and Research program (STEP) (No. 2019QZKK0603), the National Natural Science Foundation of China (No. 41971019) and the A Project Supported by Scientific Research Fund of Zhejiang University (No. XY2021014).

**Data Availability Statement:** The data presented in this study are available on request from the corresponding author. The data are not publicly available due to permissions.

**Acknowledgments:** The authors express their sincere thanks to the financial assistance from the Second Tibetan Plateau Scientific Expedition and Research program (STEP) and the National Natural Science Foundation of Chinaand the A Project Supported by Scientific Research Fund of Zhejiang University.

**Conflicts of Interest:** The authors declare no conflict of interest.

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
