# Peer review of "China’s Wealth Capital Stock Mapping via Machine Learning Methods"

_remotesensing, doi:10.3390/rs15030689_

Round 1

Reviewer 1 Report

With the support of machine learning methods, the authors conducted a comparative analysis of the spatialization methods and output results of WKS in China in 2012. On the one hand, compared with the GDP index, the WKS index is obviously more reasonable in assessing the impact of natural disasters on people's property. On the other hand, the four models currently selected by the authors (including the stacking model) are commonly used in the current machine learning researches. Through this study, the authors provided a new perspective and a new method for economic and social data gridding. Although the current research is only for the case analysis in 2012, there is no doubt that the methods and conclusions presented in this paper can be extended to 2020 or even later.

I have some small comments and suggestions.

L126: Does the road network data obtained form RESDC of the CAS represent the actual situation in 2012? Also, why did the authors only use road data and not full-feature transportation network data including roads, railways, airports, etc.? Aren't railways and airports more important in WSK evaluation?

L132: The authors clearly stated here that the POI dataset of 2010 is used. Can the author obtain the POI dataset of 2012? Compared with POI datasets, current Internet map service providers also provide ROI datasets. Have the authors considered which one is better for mapping the distribution of WKS?

L198: For datasets such as NTL/NDVI/LST daytime and nighttime, how did the author extend them from daily scale to annual scale? Please give additional explanation.

L198: Wu's paper provided all Chinese WKS data from 1978 to 2012. Why did the authors only use the data of 2012 for training and verification? What would happen if all data of all years were used for training and simulation?

L259-262: This assertion lacks evidence and requires necessary citations.

L269: Generally speaking, ensemble models can improve certain accuracy, but at the cost of more training and prediction time. From Table 1, the accuracy of the stacking model does not seem to be significantly improved. Are there any other indicators that the stacking model is better?

L301: From Table 2, none of the three basic models can approach the upper bound of the training data very well, and the same is true for the stacking model. Why are the three basic models not simulating well in the high-value zone? But the Cubist model can surpass the lower bound of training data? Perhaps the authors could have some discussion on this.

L342: Frankly speaking, in Beijing-Tianjin, the Yangtze River Delta, and the Pearl River Delta, when we compare WKS maps with land use/land cover maps or nighttime light maps, readers may think that the output of the RF model is better. Is this caused by inappropriate legend scheme? Also, why are there two legends?

No detailed information could be explored on the Chengdu-Chongqing thematic maps due to the small scale. The reviewer would suggests to replace the Chengdu-Chongqing area with  Chengdu area or Chongqing area. Also, are the scales of the 4 case areas the same? To avoid misunderstandings, it is recommended to place scale bars individually in each subfigure.

Reviewer 2 Report

This article selects RF, cubist, XGBoost, and stacking models for WKS datasets processing in mainland China. In the experimental stage, comparable results show the performances of different methods, and conclusions of suitable models are given then. The manuscript is overall well organized; however, existing careless writing should be improved through revision.  

1. The four machine learning methods are not novel, and the datasets are not sufficiently displayed. What's the contribution of this article?

2. Please provide a clearer figure and correct typos in Fig. 1 for better reading.

3. Could you further describe how to split the train and validation set for the WKS datasets?

4. On page 5, line 220, please rewrite the nomenclatures to make them consistent with eq. (3).

5. Please highlight the best result in Tables.

6. The reference number of some tables and figures are not consistent. For example, on page 7, line 270, "Figure 4 shows a ..." should be Figure 3; on page 8, line 303, "Table 4 shows the ..." should be Table 2; and on page 9, line 323, "Figure 4 shows the ..." should be Figure 5?

7. Please add corresponding descriptions about Figure 4.

Round 2

Reviewer 2 Report

The reviewer has no more comments on the current version.